# Improvement of Pursuit Eye Movement Alterations after Short Visuo-Attentional Training in ADHD

**DOI:** 10.3390/brainsci10110816

**Published:** 2020-11-04

**Authors:** Simona Caldani, Richard Delorme, Ana Moscoso, Mathilde Septier, Eric Acquaviva, Maria Pia Bucci

**Affiliations:** 1MoDyCo, UMR 7114 CNRS Université Paris Nanterre, 92001 Nanterre, France; mariapia.bucci@gmail.com; 2Pediatric Balance Evaluation Center (EFEE), ENT Department, AP-HP, Robert Debré Hospital, 75019 Paris, France; 3Child and Adolescent Psychiatry Department, Robert Debré Hospital, 75019 Paris, France; richard.delorme.rdb@gmail.com (R.D.); ana.moscoso@aphp.fr (A.M.); mathildeseptier@gmail.com (M.S.); eric.acquaviva@aphp.fr (E.A.); 4Paris 7, Paris Diderot University, 75013 Paris, France; 5Human Genetics and Cognitive Functions, Institut Pasteur, 75015 Paris, France

**Keywords:** visuo-attentional training, ADHD, children, smooth pursuit eye movements, prefrontal, fronto-striatal circuit

## Abstract

Attention-deficit/hyperactivity disorder (ADHD) is a neurodevelopmental disorder without validated and objective diagnostic procedures. Several neurological dysfunctions in the frontal circuit, in the thalamus, and in the cerebellum have been observed in subjects with ADHD. These cortical and subcortical areas are responsible for eye movement control. Therefore, studying eye movements could be a useful tool to better understand neuronal alterations in subjects with ADHD. The aim of the present study was firstly to compare the quality of pursuit eye movements in a group of 40 children with ADHD (age 8.2 ± 1.2) and in a group of 40 sex-, IQ-, age-matched typically developing (TD) children; secondly, we aimed to examine if a short visuo-attentional training could affect pursuit performances in children with ADHD. Findings showed that children with ADHD presented a greater number of catch-up saccade and lower pursuit gain compared to TD children. Differently to TD children, in children with ADHD, the number of catch-up saccades and the pursuit gain were not significantly correlated with children’s age. Furthermore, a short visuo-attentional training period can only slightly improve pursuit performance in children with ADHD, leading to a decrease of the occurrence of catch-up saccades only, albeit the effect size was small. The absence of any improvement in pursuit performance with age could be explained by the fact that the prefrontal and fronto-cerebellar circuits responsible for pursuit triggering are still immature. Pursuit eye movements can be used as a useful tool for ADHD diagnosis. However, attentional mechanisms controlled by these cortical structures could be improved by a short visuo-attentional training period. Further studies will be necessary to explore the effects of a longer visuo-attentional training period on oculomotor tasks in order to clarify how adaptive mechanisms are able to increase the attentional capabilities in children with ADHD.

## 1. Introduction

Attention deficit/hyperactivity disorder (ADHD) represents a neurodevelopmental disorder characterized by hyperactive behaviors and difficulties in controlling motor impulses and paying attention [1]. It affects up to 5% of children in the general population. Its association with frequent comorbidities leads to a major burden in childhood and later in adulthood [2]. Despite ADHD being a major issue in terms of public health, more objective approaches, preferably based on precise and valid diagnostic tools, are required [3].

Preliminary evidence suggested that some of the oculomotor abnormalities reported in ADHD could provide detailed information about the function of brain networks, particularly of the executive dysfunctions reported in these kinds of patients [4]. Children with ADHD frequently display impairments in prosaccades (saccade made in the direction of the target), in anti-saccades (saccade made in the opposite direction of the target), or in smooth pursuits. Longer latencies have also been observed during prosaccades [5], as well as an increased number of errors during anti-saccade tasks and frequent intrusive saccades during smooth pursuit in subjects with ADHD when compared to controls [6]. More recently, our group [7] reported an increased number of intrusive saccades during a fixation task, supporting previous evidence of an alteration in cognitive inhibitory mechanisms and a potential dysfunction of the frontal eye field in subjects with ADHD [8]. Remember that all these tasks (saccades and fixations) involved inhibitory control mechanisms that are mediated by the frontal eye field [9]. Both ADHD-related symptoms and saccadic eye movement impairments involved dysfunctions of pathways belonging to cortico-striato-thalamo-cortical loops, i.e., the dorsolateral prefrontal cortico-striato-thalamic-cortical network involved mainly in sustained attention [10], dorsal anterior cingulate cortico-striato-thalamic-cortical areas participating in selective attention [11], and the orbitofrontal cortico-striato-thalamic-cortical circuit related to impulsivity/compulsivity abilities [12]. Saccadic eye movement impairments also involved a more specific loop gathering the frontal eye field, the lateral intraparietal cortex, and the superior colliculi [9].

Interestingly, the training of cognitive functions, even during a short period of time, significantly affects brain networks involved in executive functions, specifically working memory, flexibility, or inhibition capabilities [13]. In subjects with ADHD, several studies have tested the effect of cognitive therapy but results were inconsistent; a recent meta-analysis that examined the efficacy of cognitive training on more specific executive functions in subjects with ADHD showed promising results [14]. By gathering the results of 22 studies in children with ADHD (age range: 3–12 years), the authors reported that cognitive training improved the domains of executive functions in children with ADHD but also led to positive and direct effects on ADHD symptoms, social skills, and academic performance. Note that the cognitive training interventions were based on games with the aim to improve specific executive functions like working memory, attention, inhibitory control, planning, and cognitive flexibility.

In our study, we compared the quality of pursuits in a group of children with ADHD and in a group of sex-, age-, intellectual quotient (IQ)-matched children with typical development. Recall that in the literature, it has been shown that the pursuit system is immature in the first years of life with a progressive maturation until adolescence [15]. Consequently, the presence of differences in cortical and central structures involved in pursuits together with the well-known abnormalities in focus attention in children with ADHD could be responsible of an eventual abnormal pattern in pursuit development in this kind of patient. We hypothesized that children with ADHD could display pursuit eye movement alteration and that this alteration could be independent of a developmental effect.

We also explored the benefit of a short visuo-attentional training on pursuit abilities in the same group of children with ADHD. Our hypothesis is that pursuit performance in children with ADHD could be sensitive to cognitive rehabilitation strategies. Note that we decided to test the effects of shorter training in order to avoid the effects of fatigue, since children with ADHD reported difficulties in focusing their attention for a long time. Recall that we just reported beneficial effects in children with dyslexia when we used short visuo-attentional training [16].

## 2. Materials and Methods

### 2.1. Subjects

In total, 40 children with ADHD and 40 sex-, IQ-, and age-matched children with TD were recruited at the Child and Adolescent Psychiatry Department, Robert Debré Hospital (Paris, France) (Table 1). Trained child clinicians evaluated all children. According to Diagnostic and Statistical Manual of Mental Disorders 5 (DSM-5) criteria [1], the diagnosis of ADHD and the main axis I comorbidities was done using the Schedule for Affective Disorders and Schizophrenia for School-Age Children-Present interview (Kiddie-SADS-EP) [17]. The ADHD Rating Scale-parental (ADHD-RS) was used to estimate the severity of the ADHD symptoms. The Wechsler scale (Wechsler Intelligence Scale for Children, fourth edition), the Beery-Buktenica Developmental Test of Visual-Motor Integration [18], and the Motor Assessment Battery for Children (MABC) [19] was used for screening all children with ADHD. Note also that all children did not receive any pharmacological treatment.

All children with typical neurodevelopment were also explored for the absence of main axis I comorbidity. To be included, they needed to have (i) ADHD-RS total raw score ≤10 [20]; (ii) a neurological examination in the normal range; and (iii) normal IQ evaluated by using the similarities test and the matrix reasoning test.

We compared pursuit eye movement performance in children with ADHD and in children with TD. To examine the effect of short training on pursuit eye movements, children with ADHD were then randomly split into two groups (G1 and G2). G1 and G2 did not differ in terms of the chronological age of the subjects and severity of ADHD measured with the ADHD-RS F(1,38) = 0.006, *p* = 0.94 and F(1.38) = 0.05, *p* = 0.87, respectively). Pursuit movements were realized two times, at T1 and T2, respectively, before and after 10 min of visuo-attentional training for the group G1, and before and after 10 min of rest (i.e., without visuo-attentional training) for the group G2. Recall that during day life, we make several eye movements to explore the natural world. However, when a subject is asked to fixate a target, attention plays a fundamental role. During the 10 min of rest, children of G2 could look everywhere, but it was not asked of them to specifically fixate on targets.

Note that we proposed the training protocol to the group of children with ADHD only since TD children reported normal behavior during pursuit eye movement performance [15].

The investigation followed the principles of the Declaration of Helsinki and was approved by our Institutional Human Experimentation Committee (Comité de Protection des Personnes CPP Île-de-France I (INSERM-CEEI-IRB, n°16-290, Hotel-Dieu Hospial). Written consent was obtained from the children’s parents after the experimental procedure was explained to them.

### 2.2. Pursuit Task

The pursuit task consisted of following a slowly moving visual target. The stimulus (a red circle of 0.5°) was moved at constant velocity (15°/s) on a 22′ computer monitor. The target was initially placed in the central position (0 deg) and then moved horizontally to one side until it reached the ±20 deg location, where it reversed abruptly and moved to the opposite side. A total of nine cycles were run and included in the analysis. Children were invited to follow the target with their eyes (for more detail, see our previous works [21]).

### 2.3. Eye Movement Recording

A non-invasive system, a medical device, the Mobile EyeBrain Tracker (www.SuriCog.com), was used to record horizontal and vertical eye movements employing an infrared camera (frequency recording of 300 Hz). The mobile EBT has a precision of about 0.25 deg. Calibration under binocular viewing was performed before pursuit recording. The calibration procedure consisted of fixating a grid of 13 points (diameter 0.5 deg) mapping the screen (for more details, see our previous works [21]).

Children were seated on a chair in a dark room, in front of a flat screen displaying the pursuit eye movements. The head of the child was held straight with a head-rest; viewing was binocular.

### 2.4. Visual-Attentional Training

We used Metrisquare© (Lebe Business Centers Sittard, Sittard, The Netherlands), a software in which different tests of visual searching were developed [22,23]. Among them, we used in our study three tasks: the house, the cat, and the space rockets. Difficulties increased from the first to the third task [23]. The total duration of visuo-attentional training was 10 min. This training consisted in three searching tasks by using Metrisquare©. The child had to search and remove small objects on a tablet with a pencil following the instructions given by the experimenter. Note that the pencil was connected with a PC, in order to obtain an objective measurement of the time needed to execute the exercise, the errors, and omission made in each exercise. The three tasks are briefly explained below. The house task corresponds to an assortment of 36 houses with similar colorful details: two blue windows, a violet roof, a violet front door, and a yellow and orange chimney. Houses have the same size, which is 2 cm length and 2 cm width. They are organized in six vertical lines and six horizontal lines. Among the houses, 12 of them have a flame coming out of the right window, 12 of them have a flame coming out of the left window, and the last 12 ones do not have any flame. The child has to cross out only the houses without flames.

The cat task consists in a set of four different drawings: a head of cats (0.5 × 1 cm), a flower (0.7 × 0.7 cm), a sunshine (0.7 × 0.7 cm), and a tree (1 × 0.5 cm). Each type is represented 24 times. The 96 drawings are randomly arranged. The child has to cross out only the cat heads.

Finally, the space rocket test represents an assemblage of 104 black and white space rockets drawings. All the space rockets have the same size: 2 cm length and 1 cm width. Space rockets are strictly organized in 13 columns and height lines. The child has to cross out the space rockets, which are identical to the given model made of two portholes, three rocket motor propellers, one ladder, and an antenna. Sixteen are identical; the 88 others have different details.

### 2.5. Pursuit Analysis

Detection of catch-up saccades during pursuit was based on criteria of minimum amplitude (2 deg) and velocity (30°/s). Catch-up saccades were saccades in the direction of the target, allowing a reduction of the position error. The number of these saccades was counted. Pursuit gain was also measured, which is the ratio between the eye velocity and the target velocity [21].

### 2.6. Statistical Analysis

We estimated the Pearson correlation (*R*^2^) between the number of catch-up saccades and age of the subject at inclusion but also the correlation between gain value of pursuit and age, in the two groups of participants (children with ADHD and children with TD) to explore the impact of age on these variables. We also ran an analysis of variance (one-way ANOVA) between two independent variables, i.e., groups of subjects (children with ADHD and children with TD), and two dependent variables, i.e., the eye movement parameters (number of saccades and the gain value). Repeated measures ANOVA was performed between the two independent variables, i.e., groups (G1 and G2) of children with ADHD on two dependent variables, i.e., the eye movement parameters (number of saccades and the gain value) recorded at the two times (T1 and T2) of the training protocol. Post hoc comparisons were made by using the Bonferroni test. We considered our results as significant when the *p*-value was below 0.05.

## 3. Results

### 3.1. Pursuit Eye Movement Performance in Subjects with ADHD and TD

A one-way ANOVA revealed that children with ADHD made significantly more catch-up saccades (F(1.78) = 36.28, *p* < 0.0001; η = 0.31) and had significantly lower pursuit gains (F(1,78) = 5.98, *p* < 0.016; η = 0.29) when compared to children with TD. In Table 2, the mean and SD values of catch-up saccades and gain during the pursuit task are shown for ADHD and TD children.

### 3.2. Impact of Development on Pursuit Eye Movement Performance

We explored the impact of age on pursuit eye movement parameters. Figure 1 reports the number of catch-up saccades in children with ADHD (A) and in children with TD (B) with respect to their age. Children with TD reported a negative correlation between the age and the number of catch-up saccades (*R*^2^ = 0.45 *p* < 0.001). In contrast, we did not observe a similar correlation for children with ADHD (*R*^2^ = 0.001 *p* = 0.83). Figure 2 shows the gain values in children with ADHD (A) and in children with TD (B). Children with TD reported a slight but significant positive correlation between the age and the gain value (*R*^2^ = 0.14 *p* < 0.016). At the opposite, children with ADHD did not display a similar correlation (*R*^2^ = 0.02 *p* = 0.36).

### 3.3. Pursuit Eye Movement Performance Recorded before and after Visuo-Attentional Training

Figure 3 shows the number of catch-up saccades at T1 and T2 for the two groups of children with ADHD (G1 and G2). We observed a significant group effect (F(1,38) = 16.78, *p* < 0.0002; η = 0.30): the number of catch-up saccades decreased significantly after the short rehabilitation program in the G1 group. The ANOVA revealed a significant interaction between training and group of children (F(1,38) =26.22, *p* < 0.00001; η = 0.40). A Bonferroni post hoc test showed that for G1, only the number of catch-up saccades decreased significantly at T2 (*p*< 0.00001).

Figure 4 represents the gain values of pursuit eye movements at T1 and T2 for the two groups of children with ADHD (G1 and G2). We did not observe any significant group effect (F(1,38) = 0.89, *p* = 0.3) (T2 vs. T1) nor any specific effect of visuo-attentional training (F(1,38) = 0.17, *p* = 0.6).

## 4. Discussion

The identification of a precise and objective diagnostic tool is a major issue in medical research, specifically in ADHD, and it is clear that the results are still rather weak overall. Through this exploratory study, we wanted to better identify the neuro-visual abnormalities of ADHD patients and identify the presence of atypical pursuit eye movement’s performance. Our goal was to identify these atypical behaviors associated with ADHD, correlated with the intensity of the disorder, not dependent on a developmental effect but sensitive to cognitive remediation. We thus compared the quality of pursuit eye movements in children with ADHD to those with TD and examined the effect of short visuo-attentional training on these movements. We observed (i) pursuit performance was not sensitive to a developmental effect in ADHD, (ii) a greater number of catch-up saccades and lower gain values in children with ADHD with respect to children with TD; and (iii) a slightly but significantly pursuit performance improvement after a short visuo-attentional training period in children with ADHD.

### 4.1. Pursuit Performance Was Not Sensitive to a Developmental Effect in ADHD

In our study, we reported a decrease in the number of catch-up saccades and an increase in the gain values in children with TD with age. Surprisingly, this developmental effect did not occur in children with ADHD. The processes of myelination, synaptic pruning, and maturation of gray matter, which reaches adult levels during adolescence, are correlated with the development of the cortical network responsible for eye movement triggering. Given the several cortical and sub-cortical structures implicated in pursuit triggering [24], the absence of a developmental effect on pursuit performances in children with ADHD could be due to a shift of maturation processes (immaturity) of the areas involved in pursuit eye movements. A major hypothesis of ADHD suggested a lag of maturation of brain structures, which is supported mainly by anatomical studies evaluating cortical thickness [25]. These anatomical studies could explain the hypothesis on the immaturity of cortical structures responsible for pursuit triggering.

### 4.2. Poor Pursuit Performance in Children with ADHD

In our study, we fostered previous findings suggesting that children with ADHD displayed visual attention abnormalities [26]. We observed that compared to age-matched typically developing children, patients with ADHD showed frequent catch-up saccades and lower gain in pursuits. In the literature, there are, however, few studies on pursuit capabilities in children with ADHD. For instance, the meta-analysis [6] reported that subjects with ADHD had more difficulties in suppressing intrusive saccades during pursuits [27,28]. More precisely, Bala et al. [27], using electro-oculography, recorded pursuit movements (catch-up saccades but not gain) in children with and without ADHD (from 5.7 to 10.4 years old). These authors found a greater number of catch-up saccades in the group of children with ADHD, compared to controls, suggesting that children with ADHD presented difficulties in moving their eyes to follow a target, most likely because of their attentional differences. Bylsma and Pivik [28] also recorded with an electro-oculography, pursuits in 20 children with ADHD and 20 typically developing children (9.6 ± 1.7 years old) and they observed that children with ADHD made more catch-up saccades with respect to typically developing children. Moreover, these authors recorded the root means square values of pursuit (which is the estimation of the cumulative distance between the eyes and the target during pursuits) and they did not report any differences between children with ADHD and those with TD. These authors suggested that subcortical dysfunctions particularly at the cerebellum level could explain the elevated number of intrusive saccades observed in children with ADHD. More recently, Gargouri-Berrechid et al. [28] recorded pursuit eye movements by using an infrared camera, at constant velocity (10°/s) in two restrictive groups of children with ADHD (*n* = 7) and matched controls. They also reported an elevated number of catch-up saccades but normal gain values in children with ADHD. The discrepancies in the gain value between this study and ours could be mainly related to the experimental set-up used, which were slightly different. Indeed, while Gargouri-Berrechid et al. [29] stimulated pursuits with a target velocity of 10°/s, in the present study, the target velocity was 15°/s. In the literature, in fact, it has been shown that the velocity of target could have an impact on the gain value [30].

In our study, we found a high occurrence of catch-up saccades and lower gain in children with ADHD compared to TD children. This could reflect an alteration in pursuit performance, given that catch-up saccades aim to maintain eye fixation on the target when the eye velocity is reduced [31]. Our findings could be explained by a difference in the ability of the prefrontal and fronto-striatal circuits to inhibit saccades during the pursuit task [32,33].

### 4.3. Effect of Visuo-Attentional Training on Eye Pursuit Performance

To further explore the relationship between abnormalities of eye pursuit movement and ADHD, we evaluated the effect of a short visuo-attentional training program on both the gain and the number of catch-up saccades during pursuits. Our results indicated that children with ADHD improved slightly but significantly their pursuit performances. After training, children with ADHD decreased significantly the number of catch-up saccades, but the pursuit gain value failed to show a significant improvement. The decreased number of catch-up saccades in ADHD during this short rehabilitation training program suggested further that catch-up saccades during the pursuit task may represent alterations during eye movement performance that are associated with attention-deficit/hyperactivity disorder, even if it is necessary to extend the duration of our training and test it on multiple different age groups of children with ADHD.

According to Strimbu and Tavel, an ideal clinical endpoint should be sensitive to an exposure or intervention, including therapeutic interventions [34]. However, our study was in its early stages and several crucial steps need to be taken to validate this objective. From our point of view, the first step will require replication of our results in a larger sample of individuals with a fine characterization of the clinical/cognitive phenotypic characteristics of ADHD patients. In particular, the relationship between executive functions and eye movement abnormalities will have to be specified. Finally, the second step will concern the therapeutic trial. The protocol for rehabilitation should be more precisely determined, and its effect on the patient’s clinical/cognitive symptoms assessed. In the future, we could propose different types of cognitive training (executive functions like working memory, attention, inhibitory control, planning, and cognitive flexibility) in order to study the impact of each of them on ADHD pathology. Moreover, we hope to be able to try a longer training period (few weeks) in order to study the effects long term and its maintenance.

## 5. Conclusions

Till now, the phenotypic measures with basic clinical information, such as sex, age, handedness, and IQ, remain the best tools to characterize patients with ADHD and to estimate the response to treatment or the prognosis [35]. Despite considerable efforts being made to identify clinical endpoints in ADHD (such as The ADHD-200 Consortium), results continue to be contrasting. The machine-assisted approach of neuroimaging/electrophysiological data may provide new directions to ADHD research and leads to the identification of patterns of clinical, cognitive, and biological features, helpful for diagnosis and treatment. Saccadic movements, which are easily measurable in patients with ADHD, could be integrated in these statistical models.

### Highlights

The effort of the scientific community to objectify the procedure of diagnosis and the prognosis or to drive the therapeutic strategies remains unfruitful in ADHD. Children with ADHD showed abnormal pursuit eye movements, specifically a greater number of catch-up saccades and lower pursuit gain.

These abnormalities appeared in our study as being independent from a developmental effect, i.e., pursuit abnormalities in children with ADHD were not correlated with the chronological age of the subjects (by contrast to children with typical development).

Finally, we observed in our study that catch up saccadic movements were very sensitive to a short visuo-attentional rehabilitation training. We observed a decreased occurrence of catch-up saccades in patients with ADHD after short cognitive remediation.

The atypical pursuit patterns in children with ADHD could represent a promising clinical feature in ADHD and could be due to the immaturity or dysfunction of central structures responsible for pursuit triggering.

## Figures and Tables

**Figure 1 brainsci-10-00816-f001:**
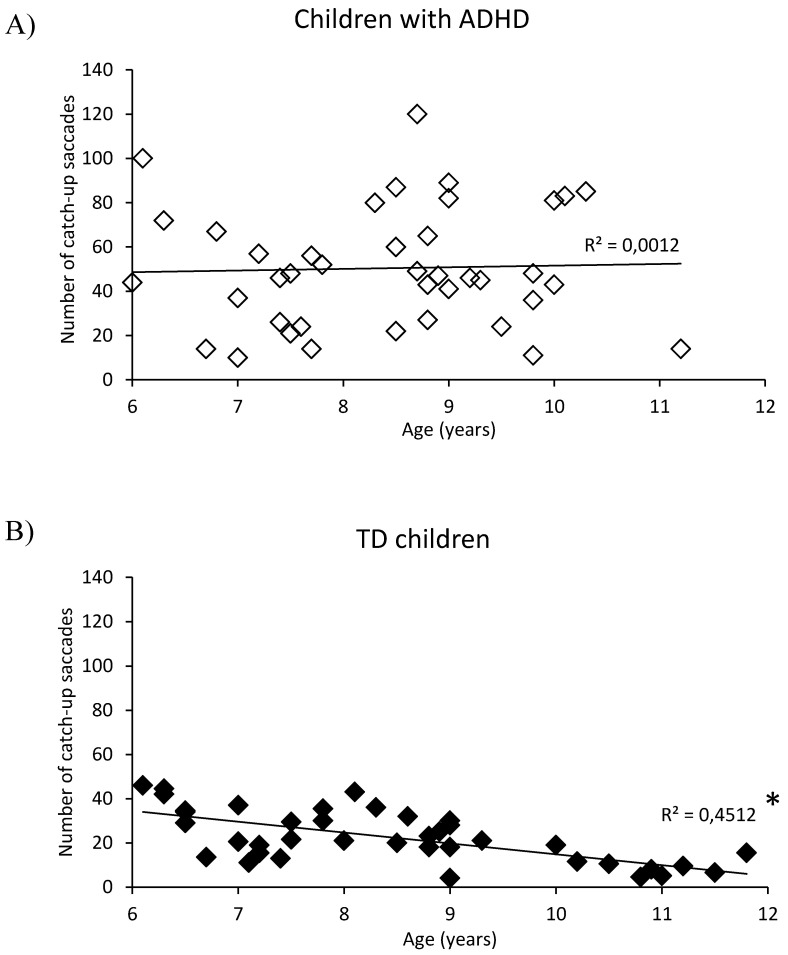
Number of catch-up saccades made by children with attention deficit hyperactive disorder (ADHD) (**A**) and with typical development (TD) (**B**) in correlation with their chronological age at inclusion. Line represents the corresponding regression by age. *: *p* < 0.05.

**Figure 2 brainsci-10-00816-f002:**
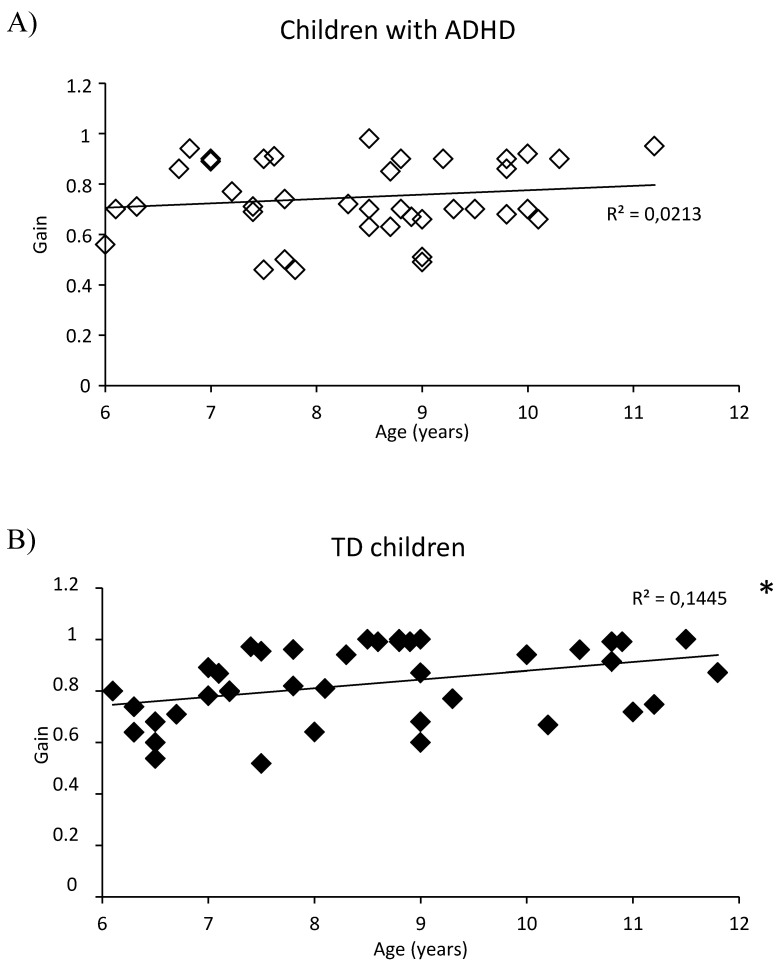
Pursuit gain made by children with ADHD (**A**) and TD children (**B**) in correlation with their chronological age at inclusion. Line represents the corresponding regression by age. *: *p* < 0.05.

**Figure 3 brainsci-10-00816-f003:**
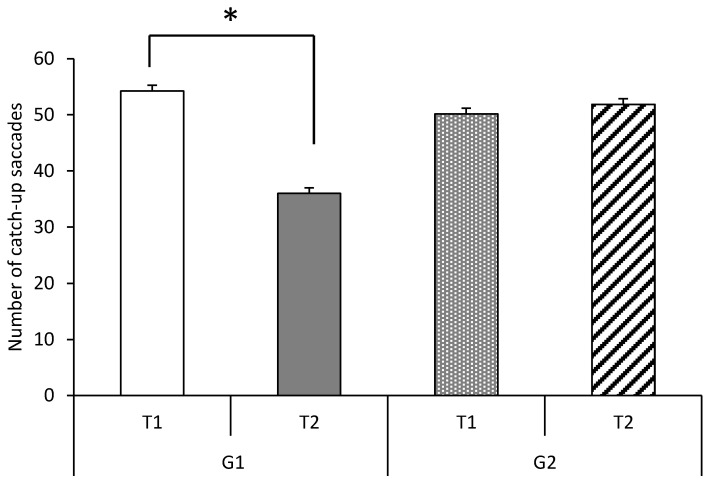
Number of catch-up saccades for children with ADHD. G1 is the interventional group and **G2**, the non-interventional group. They were both tested 2 times at inclusion (**T1**) and after the short rehabilitation training for **G1** or a resting time for **G2** (**T2**). *: *p* < 0.05.

**Figure 4 brainsci-10-00816-f004:**
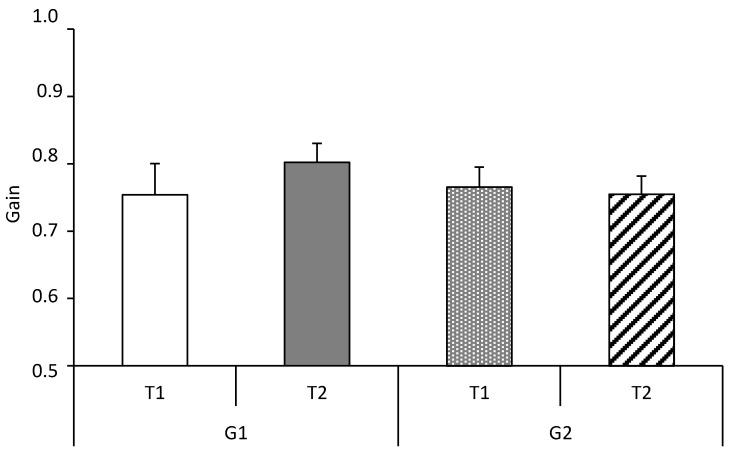
Pursuit gain values tested at **T1** and **T2** for the two groups of children with ADHD (**G1**: interventional group and **G2**: non-interventional group).

**Table 1 brainsci-10-00816-t001:** Clinical characteristics of children with typical development (TD) and with attention-deficit/hyperactivity disorder (ADHD) enrolled in the study.

	TD	ADHD
	*n* = 40	*n* = 40
**CLINICAL DATA**		
Chronological Age at inclusion (years)	8.51 ± 0.26	8.40 ± 1.32
Sex ratio M/F	25/15	28/12
**ADHD-Rating scale**		
ADHD-RS total score	3.90 ± 0.70	39.5 ± 1.7
ADHD-RS inattention sub-score	1.80 ± 0.50	19.5 ± 0.91
ADHD-RS hyperactivity/impulsivity sub-score	2.10 ± 0.80	20.0 ± 1.25
**Wechsler scale (WISC-IV) scores**		
Verbal comprehension subscale	-	99.5 ± 3.5
Perceptual reasoning subscale	-	95.6 ± 3.4
Working memory subscale	-	88.6 ± 2.9
Processing speed subscale	-	92.9 ± 2.4
Similarity total raw score	10.11 ± 0.2	9.6 ± 0.6
Matrix reasoning total raw score	10.12 ± 0.7	9.8 ± 0.5

**Table 2 brainsci-10-00816-t002:** Mean and SD values of the number of catch-up saccades and gain during pursuit for children with ADHD and TD.

	Children with ADHD	Children with TD
Number of catch-up saccades	50 ± 4	22 ± 2
Gain	0.74 ± 0.02	0.82 ± 0.02

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
