# Peer review of "Improvement of Pursuit Eye Movement Alterations after Short Visuo-Attentional Training in ADHD"

_brainsci, 2020, doi:10.3390/brainsci10110816_

Round 1

Reviewer 1 Report

I appreciated that the authors took into consideration all points raised during the revision process. The revised version of their manuscript is clearer and more suitable for publication.

Author Response

Dear editor, enclose you can find the revised ms; several words were changed/corrected in the text.

Reviewer 2 Report

Thank you for addressing my comments and questions, and for improving the paper. You have adequately addressed my concerns. 

Minor Comment :

Fig.2 : the panel A is not well align with panel B. Also, the writing R2= 0,0213 is covered, please, move up it before ".pdf" creation.

Author Response

Dear reviewer the Figure 2 was improved, according to your comments. 

Reviewer 3 Report

Authors have corrected the figures and somewhat improved the text as I requested previously. However, the MS still needs improvement both linguistically and in terms of the logical flow of the sentences. Otherwise, the readers cannot really make sense of the paper. As an example, this phrase in page 1 of introduction:

''The determinism of executive dysfunctions in ADHD, specifically response inhibition alterations, and those involved in the saccadic system impairment relies on similar brain pathways''

I guess the authors mean that:  ''The executive dysfunctions in ADHD, especially response inhibition alterations, and those involved in impairments of the saccadic system stem from similar brain pathologies''. But the phrasing is not straightforward to help the reader understand the sense of the phrase.

Author Response

Dear Reviewer according to your suggestions the sentence was corrected. 
